# Lipidomics Analysis of Human HMC3 Microglial Cells in an In Vitro Model of Metabolic Syndrome

**DOI:** 10.3390/biom14101238

**Published:** 2024-09-30

**Authors:** Mateusz Chmielarz, Mariusz Aleksander Bromke, Mateusz Olbromski, Kamila Środa-Pomianek, Magdalena Frej-Mądrzak, Piotr Dzięgiel, Beata Sobieszczańska

**Affiliations:** 1Department of Clinical Microbiology, Faculty of Medicine, Wroclaw Medical University, Chalubinskiego 4, 50-368 Wroclaw, Poland; mateusz.chmielarz@student.umw.edu.pl (M.C.); magdalena.frej-madrzak@umw.edu.pl (M.F.-M.); 2Department of Biochemistry and Immunochemistry, Faculty of Medicine, Wroclaw Medical University, Chalubinskiego 10, 50-368 Wroclaw, Poland; mariusz.bromke@umw.edu.pl; 3Department of Human Morphology and Embryology, Faculty of Medicine, Division of Histology and Embryology, Wroclaw Medical University, Chalubinskiego 6a, 50-368 Wrocław, Poland; mateusz.olbromski@umw.edu.pl (M.O.); piotr.dziegiel@umw.edu.pl (P.D.); 4Department of Biophysics and Neuroscience, Faculty of Medicine, Wroclaw Medical University, Chalubinskiego 3a, 50-368 Wroclaw, Poland; kamila.sroda-pomianek@umw.edu.pl

**Keywords:** lipidomic analysis, microglia, metabolic syndrome

## Abstract

Metabolic endotoxemia (ME) is associated with bacterial lipopolysaccharide (LPS, endotoxin) and increased levels of saturated fatty acids (SFAs) in the bloodstream, causing systemic inflammation. ME usually accompanies obesity and a diet rich in fats, especially SFAs. Numerous studies confirm the effect of ME-related endotoxin on microglial activation. Our study aimed to assess lipid metabolism and immune response in microglia pre-stimulated with TNFα (Tumor Necrosis Factor α) and then with endotoxin and palmitic acid (PA). Using ELISA, we determined cytokines IL-1β, IL-10, IL-13 (interleukin-1β, -10, -13, and TGFβ (Transforming Growth Factor β) in the culture medium from microglial cells stimulated for 24 h with TNFα and then treated with LPS (10 ng/mL) and PA (200 µM) for 24 h. HMC3 (Human Microglial Cells clone 3) cells produced negligible amounts of IL-1β, IL-10, and IL-13 after stimulation but secreted moderate levels of TGFβ. Changes in lipid metabolism accompanied changes in TREM2 (Triggering Receptor Expressed on Myeloid Cells 2) expression. HMC3 stimulation with endotoxin increased TREM2 expression, while PA treatment decreased it. Endotoxin increased ceramide levels, while PA increased triglyceride levels. These results indicated that pre-stimulation of microglia with TNFα significantly affects its interactions with LPS and PA and modulates lipid metabolism, which may lead to microglial activation silencing and neurodegeneration.

## 1. Introduction

Lipids, the primary and essential biomolecules of the brain, constitute 50% of its dry weight; hence, their abundance in the central nervous system (CNS) underscores their significant role in various physiological processes in the brain, e.g., synaptogenesis and neurogenesis. Lipids play a vital role in the cell membrane signal transduction, a crucial process for brain function. In addition to this role and their involvement in cell membrane structure, lipids also serve as energy substrates, providing 20% of the brain’s energy from their oxidation [1,2,3]. The brain has a unique lipid composition, mainly cholesterol and sphingolipids. While sterols and SFAs are synthesized in the brain de novo, the majority of long-chain polyunsaturated fatty acids (LC-PUFAs), such as arachidonic acid, eicosapentaenoic acid, and docosahexaenoic acid, are not produced in the brain and must be obtained from the diet [3,4]. This emphasizes the importance of a diet rich in unsaturated fatty acids for brain health. The harmful effects of a high-fat diet (HFD) on human health have been known for a long time, and these effects extend to the CNS, leading to endoplasmic reticulum stress and mitochondrial dysfunction in microglia and neuronal cells. A deep body of literature has highlighted the damaging impact of HFD on the human brain microglia, with long-term HFD carrying the risk of memory disorders and cognitive and emotional dysfunctions related to changes in the lipid profile. Even short-term exposure of the CNS to postprandial SFAs circulating in the blood that reaches the brain induces significant changes in microglia and can promote gliosis in the hypothalamus. Dietary lipids are rapidly sent to the brain via passive diffusion of free fatty acids bound to plasma albumin, a transcytosis pathway involving low-density lipoprotein (LDL) receptors and clathrin- and caveolae-dependent endocytosis, and finally, the involvement of lipoprotein receptors after the re-esterification of fatty acids into a glycerol backbone, i.e., lysophosphatidylcholine (lysoPC) [5]. In obese individuals, HFD intake leads to elevated levels of SFAs, specifically PA in plasma, which increases the transport and accumulation of these lipids in the brain [6].

Microglial cells are vital in maintaining the proper physiological state of the brain via controlling neuronal functioning, maturation, and the formation of new connections [7,8,9,10,11]. Although the primary function of microglia is to protect the nervous system, its overactivation may lead to neurotoxic effects. Under physiological conditions, microglia survey and constantly monitor their microenvironment, neuronal, and synaptic activity, and so they present the Homeostasis-Associated Microglia (HAM) phenotype. In contrast, contact of microglial cells with factors crossing the blood-brain barrier (BBB) or environmental cues switch microglia to an activated state referred to as Disease-Associated Microglia (DAM) [12,13,14]. Furthermore, DAM microglia based on the scRNAseq of distinct neuronal immune cells under pathological conditions allowed the distinction of a DAM subtype, i.e., the Neurodegeneration-Associated Microglia (NAM) [15,16]. DAM produces various pro-inflammatory cytokines, reactive oxygen species (ROS), and glutamic acid, resulting in an inflammatory response at the site of injury, the degradation of the cell matrix, destabilization of synaptic connections, and promotion of retraction of dystrophic neurons [11,13,17,18]. Additionally, DAM presents enhanced lipid metabolism, characterized by increased expression of lipid-related genes, including Apolipoprotein E (ApoE), Triggering Receptor Expressed on Myeloid Cells 2 (TREM2), and Lipoprotein Lipase (LPL) [14]. ApoE and TREM2 are both involved in brain neurodegeneration, linking neuropathology in the CNS to lipid metabolism. TREM2 is a transmembrane receptor exclusively expressed in the brain in microglia, acting as an extracellular lipid sensor that enhances microglia activity and is involved in neurological disorders like Alzheimer’s disease (AD) [19].

ME, related to bacterial endotoxin leaking from the gut into the bloodstream and increased levels of SFAs, leads to systemic subacute inflammation. This condition usually accompanies obesity and HFD. Lasting for years, ME can ultimately lead to changes in the brain, indirectly through markers of subacute inflammation or directly through the penetration of endotoxin and increased amounts of SFAs into the brain. Numerous studies confirmed the effect of endotoxemia associated with ME on microglial activation [20].

Despite the significant impact of lipids on brain function, there is surprisingly little research on lipid metabolism in microglia in ME, which affects most obese individuals. With the above in mind, our in vitro studies concentrated on the lipid profile of HMC3 microglial cells pre-stimulated with the pro-inflammatory cytokine TNFα to create conditions resembling systemic inflammation. In our ME model in vitro, activated microglial cells were treated with endotoxin at clinically irrelevant concentrations and high levels of SFAs represented by palmitic acid (PA), i.e., in conditions corresponding to ME in vivo.

## 2. Materials and Methods

### 2.1. Chemicals Used in the Study

PA and LPS from *E. coli* O55:B5 were obtained from Sigma-Aldrich (Darmstadt, Germany), and fatty acids-free bovine serum albumin (BSA) was obtained from BioWest (49340 Nuaillé, France). Cell culture, fetal bovine serum (FBS), media, and supplements were all purchased at ThermoFisher Science. ELISA kits for IL-10, IL-13, TGFβ, and IL-1β were from ThermoFisher Science (Waltham, MA, USA). All chemicals and kits for real-time PCR were obtained from the Bio-Rad (Warsaw, Poland).

### 2.2. Microglial Cells Culture

The study was performed on commercially available human microglial cells of the HMC3 cell line (purchased from ATCC (LGC Standards, Łomianki, Poland) collection CRL-3304^TM^) cultured in EMEM medium supplemented with 10% FBS and penicillin-streptomycin (10,000 U and 10 mg per mL, respectively) at 37 °C in 5% CO_2_ in a humid atmosphere. The HMC3 cell line was established from human embryonic microglial cells by SV–40 immortalization and authenticated by the American Type Culture Collection (ATCC). Regarding markers and functions, especially inflammatory responses and phagocytosis, HMC3 cells resemble primary microglia [20]. Routinely, HMC3 cells were cultured in standard 75 mL cell culture-treated flasks with a filter cap (Greiner Bio-One, Kremsmünster, Austria). After purchase, the cells were expanded, and the second and third passages were frozen in liquid nitrogen. For experiments, the cells were thawed and used after two or three passages and expansion. Usually, the cells formed a monolayer within 72–96 h. All studies used cells between the sixth and ninth passages, which reached about 80% monolayer. Culturing HMC3 cells to a complete monolayer caused morphological changes, gradual detachment of the cells, and decreased viability. For the study, HMC3 cells were stimulated with 25 ng/mL TNFα for 24 h to activate microglial cells and then treated with PA at 200 µM and LPS at 10 ng/mL or both for 24 h.

### 2.3. Palmitic Acid Preparation

PA was dissolved to the concentration of 500 mM in NaCl and heated at 70 °C to obtain a clear solution. Fatty acids-free and endotoxin-free BSA was dissolved in an EMEM medium to receive a 10% solution and filter-sterilized. Then, PA was combined with BSA, vortexed, and sonicated for 15 min at 55 °C twice. The final concentration of PA in BSA was 5 mM, and the PA to BSA ratio was 3.2:1. Next, the PA-BSA solution was split into small portions and frozen at 20 °C. A 10% BSA-NaCl solution was prepared as vehicle control. The PA solution was heated at 55 °C for 15 min before dilution to the appropriate concentration (200 µM) in the cell culture medium for the experiments.

### 2.4. Experimental Conditions

All experiments were performed on HMC3 cells pre-stimulated for 24 h with 25 ng/mL TNFα and then treated for 24 h with PA (200 µM), LPS (10 ng/mL), or both PA and LPS diluted in a cell-culture medium. TNFα-pre-stimulated but untreated with LPA or PA HMC3 cells and pre-stimulated cells treated with PA solvent, i.e., BSA-NaCl, served as the negative and the vehicle controls, respectively.

### 2.5. Microglial Cell Viability and Morphology

The cell viability and proliferation were measured with MTT (3-[4,5-dimethylthiazol-2-yl]-2,5-diphenyl) tetrazolium bromide diluted in PBS to obtain a stock solution of 5 mg/mL which was further diluted to 0.5 mg/mL in the cell culture medium. The HMC3 cells were seeded at 5 × 10^4^ cells/mL into a 96-well cell culture plate and incubated overnight to obtain an 80% monolayer. After stimulation with TNFα and treatment with PA, LPS, and PA+LPS as described above, the culture medium was removed, 100 µL of MTT solution per well was added, and the plate was incubated for 3 h at 37 °C in a 5% CO_2_ atmosphere. MTT is metabolized by living cells to form water-insoluble purple formazan crystals, which dissolve in DMSO to form a colored solution. The color intensity correlates with cell viability. The resulting colored reaction was quantified by measuring absorbance at 570 nm in a multi-well spectrophotometer. The assay was performed twice in eight wells each. HMC3 cell morphology was assessed by fluorescent green staining of the cellular actin cytoskeleton with FITC (Fluorescein IsoThioCyanate), Hamburg, Germany-conjugated phalloidin. Cells were cultured on sterile glass slides (13 mm in diameter) placed in the wells of a 24–well cell culture plate. After stimulation with TNFα and treatment with LPS, PA, and LPS+PA as described above, the cell culture media were removed, and cells were washed three times with pre-warmed PBS and fixed for 10 min with a 4% buffered formalin solution. After fixation, the cells were washed repeatedly with PBS solution to remove residual formalin. Then, the cell membranes were permeabilized with 0.5% Triton X-100 solution in PBS for 5 min to allow the penetration of phalloidin-FITC into the cells. After that, the cells were washed several times and stained with phalloidin (5 µL stock of phalloidin-FITC in methanol diluted in 200 µL PBS) for 20 min. In parallel, the cell nuclei were stained with DAPI dye (working concentration 1 mg/mL in PBS), which exhibits blue fluorescence after binding to nuclear DNA. After incubation, the dyes were washed off with sterile distilled water; the slides were removed from the wells onto glass slides, and after drying, they were viewed under a fluorescence microscope (Olympus BX-51, Hamburg, Germany). The cell morphologies were documented with photographs.

### 2.6. Quantitative Real-Time for Trem2 Expression Assay

Total RNA was isolated using the Aurum TM Total RNA kit (Bio-Rad) according to the manufacturer’s instruction and transcribed to cDNA with the iScriptTM—Reverse Transcription Supermix for RT-qPCR. The primers for Trem2 used were as follows: forward: 5’ TCT GAG AGC TTC GAG GAT GC 3’ and reverse: 5’ GGG GAT TTC TCC TTC CAA GA 3’. The reaction was performed in 10 μL volumes using the Sso AdvancedTM Universal SYBR^®^ Green Supermix on a MIC Real-Time PCR System. RT-PCR reactions have run in triplicate in the following conditions: activation of the polymerase at 95 °C for 15 min, initial denaturation at 95 °C for 15 sec, annealing at 60 °C for 20 sec, and elongation at 72 °C for 20 sec followed by 45 cycles. The relative mRNA expression of the gene was normalized against the reference GAPDH gene and calculated with the ∆∆Ct method according to the protocol by Livak and Schmittgen [21].

### 2.7. Lipidomic Analysis

Three biological replications of HMC3 cultures were grown under the standard experimental conditions described above for the lipidome analysis. Each monolayer was scraped, and cells were collected, gently centrifuged, and washed with PBS before the extraction. In the next step, cells were quenched. Lipids from the cell pellets were extracted with the use of 1000 µL of cold (–20 °C) mixture of methyl-tert-butyl-ether: methanol (3:1, *v*/*v*) with the addition of internal standards (0.1 µg/mL deuterated phosphatidylcholine (PC36:0-D70) and 0.1 µg/mL deuterated arachidonic acid (AA-D5)). The samples were then sonicated using a cooled (4 °C) ultrasonic bath for 10 min. Then, a 500 µL mixture of water and methanol (3:1, *v*/*v*) was added to each sample, forming two liquid phases—polar and nonpolar. The lipid-containing nonpolar phase was collected, dried with a speed vac, and stored at –20 °C before the lipidomic profiling.

The nonpolar phase lipids were analyzed using a Waters Acquity UPLC system coupled with a Xevo G2 QTof mass spectrometer (Waters, Milford, CT, USA). The LC conditions were column ACQUITY UPLC BEH Shield RP18 (2.1 × 100 mm, 1.7 µm, Waters); column temp. 60 °C; mobile phase A acetonitrile/water (60:40, *v*/*v*) with 10 mM ammonium formate and 0.1% formic acid; mobile phase B isopropanol/acetonitrile (90:10, *v*/*v*) with 10 mM ammonium formate and 0.1% formic acid; injection volume 2.5 µL. A constant flow rate of 300 µL/min was maintained over 25.5 min of an analytical run. Following mobile phase gradient was used: initial 70.0% A, 3.0 min. 70% A, 6.5 min. 55% A, 12.5 min. 40% A, 19.0 min. 5% A, 21.5 min. 5% A, 21.51 min. 70% A, 25.5 min. 70%. The mass spectrometry conditions were: ionization ESI positive, capillary voltage 2.8 kV (positive), source temp. 120 °C, desolvation temperature 450 °C, acquisition mode MSE (low CE 10V, high CE 30V). From each sample, 5 µL was injected in three technical replicates.

Raw-acquired chromatograms were converted to .mzXML files by MSConvert (a part of ProteoWizard package, version 3.0.23051) [22]. For peak detection, alignment, annotation, and peak area integration, the files were loaded into the XCMS server (https://xcmsonline.scripps.edu, accessed 31 May 2024). The output file was further compared to deuterated standards (Ultimate SPLASH ONE mix, Avanti Research, Birmingham, AL, USA) and an in-house database of lipids for refined manual annotation of peaks. All annotated peaks were normalized to internal standard areas and scaled to the control mean. Bar plots present the mean (*n* = 3) value of scaled data. One-way ANOVA and Tukey’s HSD test (R package ‘stats’ version 4.5.0) were performed to detect statistically significant differences between treatments. Heatmaps display the mean area scaled to the analyte’s mean value across the experiment and log2-transformed. Hierarchical clustering based on Euclidean distance was applied to form clusters of analytes and profiles (R package ‘gplots’ version 3.1.3). The calculations and plotting were performed using Excel (Microsoft Office 2021) and R (R version 4.3.1; R package ‘corto’ version 1.2.2).

### 2.8. Cytokine Production Assays

IL-1β, IL-10, IL-13, and TGFβ levels were assessed in a cell culture media collected from HMC3 cells using commercial immuno-enzymatic kits. According to the manufacturer’s instructions, the assays were performed in triplicate with undiluted cell culture medium samples. The assay range was 3.15–200 pg/mL for IL-10 ((ThermoFisher Science; Waltham, MA, USA; catalog #BMS215-2), 1.6–100 pg/mL for IL-13 ((ThermoFisher Science; Waltham, MA, USA; catalog#BMS231-3), 3.9–250 pg/mL for IL-1β ((ThermoFisher Science; Waltham, MA, USA; catalog#BMS224-2), and 31–2,000 pg/mL for TGFβ ((ThermoFisher Science; Waltham, MA, USA; catalog#BMS249-4).

### 2.9. Statistical Analysis

All experiments on HMC3 cells were performed independently three times in duplicate. A *t*-test was used to determine the differences between the means of the two groups. The one-way analysis of variance (ANOVA) was used to determine the differences among cells treated with LPS, PA, or both, with a *p*-value < 0.05 considered statistically significant. All data are presented as mean ± standard deviation.

## 3. Results

### 3.1. Palmitic Acid but Not Endotoxin Decreases the Viability of Activated HMC3 Cells

LPS at 10 ng/mL did not affect HMC3 cell viability compared to untreated control. In contrast, PA and PA combined with LPS decreased TNFα-activated microglial cell viability by ca. 25% (*p* < 0.05) compared to the vehicle control and LPS (Figure 1).

### 3.2. Microglial Cell Morphology

Unstimulated HMC3 cells showed slightly elongated morphology with well-visible actin fibers (Figure 2A). In contrast, cells treated with TNFα for 24 h appeared rounded with an actin accumulation at the cell’s edges (Figure 2B). TNFα-stimulated cells treated with vehicle control were more elongated and protruded (Figure 2C), whereas cells treated with LPS showed higher actin fiber tension and single round cells (Figure 2D). Treatment of pre-stimulated HMC3 cells with TNFα and then PA again changed the morphology-cells were generally elongated and presented wide pseudopodia (Figure 2E). The effect of treating pre-stimulated cells with PA and LPS was an increase in the number of rounded cells, specifically for microglia with increased phagocytosis (Figure 2F).

### 3.3. Cytokine Profile in HMC3 Cells

The HMC3 cells activated with TNFα cells did not produce detectable amounts or secreted negligible amounts of the pro-inflammatory cytokine IL-1β and anti-inflammatory cytokines IL-10 or IL-13 in contrast to TGFβ (Figure 3A). Negative control cells secreted TGFβ at approximately 60 pg/mL. Vehicle controls produced slightly higher levels of TGFβ (72.3 pg/mL; *p* > 0.05 compared to negative control). In turn, cells treated with LPS, PA, and these two compounds produced significantly higher levels of TGFβ than controls (91.8 pg/mL, 81.4 pg/mL, and 110.5 pg/mL, respectively; *p* < 0.05). Interestingly, the highest TGFβ levels produced HMC3 cells treated with LPS and PA, indicating their mutually enhancing anti-inflammatory effect. These results suggest that long-term pre-stimulation of microglial cells with pro-inflammatory signals induces an anti-inflammatory response or immune tolerance in microglia after 24 h.

### 3.4. LPS Increases Trem2 Expression in Microglial Cells

The observed changes in the lipid profile in microglial cells treated with PA and LPS suggested the involvement of receptors sensing dietary lipids. Hence, in this study, we also assessed the transcription of the *Trem2* gene as an extracellular lipid sensor exclusively expressed in microglia (Figure 3B). The vehicle control did not affect *Trem2* expression compared to negative control, while LPS increased its expression by 0.75 times (*p* =0.001). In turn, PA slightly increased the expression of *Trem2* by 0.25-fold (*p* > 0.05), while the combination of LPS and PA reduced the expression of this gene, although non-significantly (*p* > 0.05), which indicates the attenuation of the LPS-induced increased expression of *Trem2* by PA in the experiment conditions.

### 3.5. Lipidomic Analysis

The lipidomic profiling of HMC3 cells provided us with data on levels of 1483 analytes. Of this number, we were able to identify 146 lipid species. The general picture of the changes in lipids is visualized with the heatmap in Figure 4. The heatmap presents identified lipids expressed as log_2_ transformed ratios between the treatment mean and the control. Our experimental conditions resulted in profound changes in the lipidome of HMC3 cells. The treatment with PA (with or without LPS) resulted in a distinguished profile characterized by increased relative concentration of many triacylglycerides (TAG) in HMC3 cells. The highest increase was more than eight-fold above the control level. On the other hand, in LPS-alone treated cells, a relatively small reduction of TAG concentration was observed. In the presence of PA, a decrease in the concentration of some nitrogen-containing lipids, such as phosphatidylcholine (PC), phosphatidylethanolamine (PE), and sphingomyelin (SM), was observed. In contrast, ceramides were increased in LPS-only treated cells. A heatmap of all found analytes is available in the Appendix A.

The following figures present statistically significant changes in three groups of lipids: membrane phospholipids (Figure 5), sphingolipids and ceramides (Figure 6), and triacylglycerols (Figure 7) of microglial HMC3 cells pre-stimulated with TNFα and then treated with PA and LPS. Within the group of bilayer phospholipids fulfilling the role in cellular metabolism and signal transduction, PA alone significantly increased the abundance of phospholipid PC32:0 (PC16:0/16:0) and phosphatidylethanolamine PE34:0 (PE18:0/16:0) compared to controls. The same effect was observed for the combination of PA with LPS and PC32:0. However, LPS alone significantly decreased their abundance. LPS also reduced the level of PC34:0 (PC18:0/16:0) phosphatidylcholine. A decrease in phosphatidylcholine PC36:2 (PC18:1/18:1) abundance was also observed when cells were treated with PA alone or combined with LPS. Furthermore, PA separately or in combination with LPS decreased the abundance of sphingomyelin SM(d18:1_14:0) and SM(d18:1_24:1), while LPS alone increased these lipids in microglia relative to untreated controls. In pre-stimulated HMC3 cells treated with PA individually or combined with LPS, the abundance of two triacylglycerols, i.e., TAG48:0 (tripalmitate glycerol 16:0/16:0/16:0) and TAG60:11 (20:5/20:5/20:1), increased. LPS alone induced a decrease in levels of TAG60:2 and TAG60:3 compared to controls. The abundance-lowering effect was also observed when HMC3 cells were treated with PA combined with LPS. At least, PA alone decreased the abundance of some ceramides in HMC3 cells, i.e., Cer(d18:1/24:0), Cer(d18:1/24:1), and Cer(d18:1_16:0) compared to vehicle control and cells treated with LPS which increased their level. Moreover, LPS increased the abundance of Cer(d18:1_22:0) relative to the negative control.

## 4. Discussion

Obesity and ME are accompanied by systemic subacute inflammation and associated pro-inflammatory cytokines, which can activate microglia. The study focused on lipid metabolism in activated human microglial cells stimulated with PA and LPS, the two most pro-inflammatory challenges in ME. Both these elements activate cellular inflammatory responses via different toll-like receptors (TLRs). PA activates TLR2, whereas LPS activates TLR4-associated pathways, thus determining different cellular inflammatory reactions, which may critically impact neurodegeneration [23,24]. Several studies have reported differential activation of microglia in response to free fatty acids and LPS [25,26,27]. Kappe et al. [27] demonstrated that LPS and PA treatment of mouse microglial BV-2 cells induced opposite effects, where palmitate reduced the mRNA expression and secretion of TNFα, whose induction is characteristic of LPS activation. Other studies have also reported differential macrophage and microglia activation in response to endotoxin and PA [25,26].

In the study for microglia activation, we used TNFα, which level increases rapidly in the blood during endotoxemia [28]. Moreover, TNFα crosses the intact BBB by receptor-mediated transport [29] and enhances TLR2 expression in mouse primary microglia [30]. Additionally, exogenous blood-derived TNFα via its receptor TNFR1 (Tumor Necrosis Factor Receptor 1) induces in microglia the production of TNFα and other inflammatory mediators, i.e., pro-inflammatory cytokines, nitric oxide (NO), and free radicals (ROS), through a positive feedback loop, maintaining prolonged microglial activation [31]. TNFα is a pleiotropic cytokine that regulates different homeostatic brain functions, such as synaptic plasticity, blood–brain barrier (BBB) permeability, myelination, glial transmission, as well as cognitive abilities (learning and memory), but also sleep, water/food intake. On the other hand, TNFα can exert a neurotoxic effect, as demonstrated in AD and Parkinson’s disease (PD), epilepsy, multiple sclerosis (MS), as well as chronic pain, ischemic and traumatic brain injury, and infectious diseases, among others [32,33]. In a normal state, non-activated microglia release low fluctuating levels of TNFα, which affect astrocytes, oligodendrocytes, and neurons, inducing the release of moderate levels of glutamate by astrocytes that modulate synaptic activity [33,34]. However, in brain inflammation, activated microglia release much higher TNFα levels, which, via the astrocyte receptor TNFR1, stimulate the production of prostaglandins and massive release of glutamate, altering synaptic function and causing neuronal damage [33]. TNFα release by microglia also affects microglial functions controlling the expression of protective factors, i.e., granulocyte-colony stimulating factor (G-CSF), IL-10 via TNFR2, regulating peripheral immune cells recruitment, and sustaining cytokine release via TNFR1 [34].

First, our study searched for chosen cytokines in culture media to establish a microglia activation state upon TNFα-stimulation and treatment with PA and LPS. The results showed that HMC3 cells produced both pro-inflammatory cytokines, like IL-1β, and anti-inflammatory cytokines, IL-10 and IL-13, at marginal levels. Similarly, Kong et al. [35] demonstrated that treating BV2 cells with TNFα did not induce IL-1β production. Interestingly, subsequent HMC3 cell stimulation with LPS or PA also did not increase these cytokine releases, suggesting a lack of microglial activation. On the other hand, Chen et al. [36] in their study stimulated HMC3 cells with LPS at a dose of 250 ng/mL and showed an increase in the secretion of various pro-inflammatory cytokines and chemokines, including IL-1β, the level of which increased ca. 17-fold, confirming the ability of these cells to synthesize and secrete the cytokine. This clearly indicates that divergent research results from different authors regarding the detectability of IL-1β produced by HMC3 cells may depend on many factors, such as cell activation states, culture conditions, cell passage number, the time elapsed between sample collection and processing, and the storage conditions. This variability can influence the accuracy of the IL-1β measurements, leading to instances where IL-1β levels fall below the detection threshold, similar to our results.

The HMC3 cell line produces negligible levels of IL-10 [37], which corresponds with our results, but can secrete IL-13. Caruso et al. [38] demonstrated that IL-13 in HMC3 cells treated with amyloid β (Aβ42) was upregulated shortly after exposure to Aβ42 but downregulated after prolonged incubation. Lacavalla et al. [39] reported that LPS-activated HMC3 cells produced IL-13 levels below the detection limit, consistent with our research results. Similarly, according to Pallio et al. [40], HMC3 cells challenged with IL-1β showed a significant increase in the expression and production of pro-inflammatory cytokines TNFα and IL-6 but a decrease in the anti-inflammatory IL-13 cytokine. In the study of Wang et al. [37], stimulation of HMC3 cells with TNFα (300 ng/mL, 24 h), although increased the percentage of cells expressing the markers of inflammatory microglia, i.e., CD14, IBA-1, CD-40, did not stimulate the production of anti-inflammatory IL-10 and IL-4 cytokines and anti-inflammatory surface markers, i.e., CD163 and CD206. All this indicates that the stimulation of microglia by LPS and PA largely depends on the pro-inflammatory milieu affecting microglia in different ways.

Lively et al. [41] compared the effects of LPS and TNFα+INFγ on primary rat microglial cells and demonstrated that pro-inflammatory cytokines and LPS had a different impact on microglial stimulation. Among others, LPS significantly upregulated the IL-1β gene and its receptor (IL-1R1), whereas both cytokines increased the IL-1 receptor antagonist (*il-1rn*), keeping microglial cells less responsive and thus attenuating the LPS effect. In their study, TNFα+INFγ decreased microglia responsiveness by increasing the CCR5 receptor, a proposed decoy receptor for the pro-inflammatory chemokines CCL3, CCL4, and CCL5, while LPS reduced the expression of these chemokines. In addition, TNFα+INFγ reduced microglial TLR4 expression. These data indicate that although LPS and TNFα+INFγ are all pro-inflammatory stimulants, they often have opposing effects on the expression of inflammatory modulators. Hence, the prior activation of microglia with TNFα following LPS and PA treatment, used in our studies, could switch microglial cells to less responsive to pro-inflammatory signals, protecting the microglial overactivation, confirmed by TGFβ secretion.

In the study, the TNFα-activated HMC3 cells treated with LPS and PA produced moderate TGFβ levels. TGFβ, although classified as an anti-inflammatory cytokine, presents a pleiotropic effect by activation of immune cells, inflammatory response, and damage repair. Hence, TGFβ is involved in initiating and resolving inflammation. Its ability to down-regulate pro-inflammatory cytokines is essential to its function in healing processes [16,42,43]. The TGFβ1, expressed endogenously in microglia, astrocytes, endothelial cells, and choroid plexus cells, seems to be crucial to protecting microglia from excessive activation [16]. Hence, an increase in TGFβ1 levels is considered neuroprotective by down-regulation of pro-inflammatory cytokines, thus limiting inflammation, reducing the accumulation of lipid droplets (LDs) in microglia, decreasing amyloid β (Aβ) deposition, and clearing amyloid deposits [3,16,44,45]. In an animal model, Mendes et al. [46] demonstrated that consumption of SFAs increases hypothalamic TGFβ1 expression, attenuating inflammation, improving energy metabolism, and protecting animals from obesity. On the other hand, TGFβ1 is also involved in neurodegeneration [42,47]. In AD patients, the level of TGFβ1 in plasma and CSF is increased, which correlates with reduced expression of the TGFβ1R2 receptor [48]. Furthermore, it has been shown that loss of TGFβ signaling in the microglia resulted in motor deficits and impaired myelination by disturbances in oligodendrocyte maturation [16,49]. According to Yang et al. [3], the increase in TGFβ1 levels may be compensatory, balancing the reduced expression of TGFβ1R2. In our research, LPS and PA separately and together increased TGFβ levels in activated microglia, suggesting that the pro-inflammatory challenge induced an anti-inflammatory response to reduce microglial activation. On the other hand, research by Mitchell et al. [50] showed that LPS at 10-100 ng/mL reduced the expression of TGFβ receptors, i.e., TGFβR1 and TGFβR2, and the level of Smad2 protein, a key mediator of TGFβ signaling pathway. Additionally, LPS antagonized the ability of TGFβ to suppress the expression of pro-inflammatory cytokines, which may ultimately contribute to neurodegeneration. Hence, it is tempting to speculate that ME, associated with the constant presence of endotoxin and SFAs in the blood and chronic systemic inflammation, can induce alternative activation in microglia, ultimately switching it towards the NAM phenotype. However, more studies must be performed on these mechanisms and co-interactions between endotoxin and TGFβ signaling.

Interestingly, in our study, the increased TGFβ production was accompanied by decreased *Trem2* expression in HMC3 cells treated with PA combined with LPS. TREM2 is a glycoprotein receptor in the cell membrane that participates in signaling through association with the adapter protein DAP12 (Death-Associated protein 12) [51]. TREM2 broadly regulates phagocytosis, autophagy, and cytoskeletal remodeling in microglia and, more importantly, is involved in lipid metabolism. Hence, our study also examined TREM2 expression as a crucial microglial receptor engaged in taking up glycolipids from the membranes of degraded neuronal processes and myelin debris [52,53,54].

The results demonstrated that LPS in HMC3 cells pre-stimulated with TNFα enhanced *Trem2* expression, contradicting the results of Piccio et al. [55], in which cultured microglia treated with LPS at a concentration of 1 mg/mL for 30 min completely down-regulated the expression of *Trem2*. Similarly, in an animal model, Owens et al. [56] showed that *Trem2* expression in microglia is inhibited by approximately 90% in response to endotoxin. However, in these previously described in vitro and in vivo research models, the effect of LPS on *Trem2* expression was tested after much shorter incubation time, with much higher LPS doses and without pre-stimulation of microglial cells with pro-inflammatory cytokines, so they are entirely different from the conditions used in this study. Thus, our results indicated that TNFα-activated microglial cells respond differently concerning TREM2 expression than primary, unstimulated microglia.

Moreover, in our research, PA did not affect *Trem2* expression, unlike LPS. In contrast, LPS and PA simultaneously decreased *Trem2* expression, suggesting the opposing effects of PA and LPS on *Trem2* expression. TREM2 expression decreases during acute inflammation but increases during resolution of inflammation and in mouse models of brain amyloidosis and Tau pathology [37]. Hence, the ability of LPS at low concentrations to increase TREM2 in pro-inflammatory stimulated microglia suggests the anti-inflammatory effect of low endotoxemia, possibly due to immune tolerance or decreased microglia responsiveness caused by TNFα stimulation. On the other hand, PA combined with LPS significantly decreased TREM2 expression, indicating that PA in combination with LPS can stimulate the DAM phenotype of microglia. The decrease in *Trem2* expression by PA may also indicate a protective effect on microglial cells against excessive lipid transport. However, if microglial cells can regulate the expression level of lipid receptors in response to lipid influx, this means that LPS, by increasing TREM2 levels, may increase lipid uptake by cells. However, this hypothesis requires detailed studies.

The taken-up glycolipids in microglia are subject to lysosomal degradation [57]. Hence, our further studies focused on lipidomics analysis to investigate the impact of altered TREM2 expression on lipid metabolism in microglia. The results showed that the pro-inflammatory challenge of HMC3 cells caused significant metabolic changes in three lipid groups: sphingolipids, membrane phospholipids, and triacylglycerols (TAGs).

Sphingolipids (SPs) are bioactive lipids of neuronal cell membranes and play a crucial role in metabolism, brain development, and health; they are involved in myelin stability and neuron–glial connections and are related to neuronal differentiation and synaptic transmission. SPs include ceramides, sphingomyelins, and Sph-1-phosphate (S1P), which regulate cell growth, differentiation, senescence, and apoptosis. Sphingolipid metabolism is associated with various neurological disorders [3]. Ceramides, as vital membrane bioactive sphingolipids, play critical roles in the brain, modulating many neuronal and glial processes such as apoptosis, cell differentiation, astrocyte activation, and inflammatory reactions; hence, altering the balance of different ceramides may contribute to various diseases, including neurodegenerative diseases, e.g., AD and PD [58,59]. There is a growing body of evidence pointing to the critical role of ceramides in stroke and its effects, e.g., depression and cerebral small vessel disease (CSVD) [60]. In microglia, ceramides, especially long-chain ceramides (C16-24), activate pro-inflammatory pathways via transcription factor NF-ĸB regulating the release of pro-inflammatory cytokines, such as eicosanoids, as well as cell apoptosis and numerous receptor-mediated pathways [58,59]. Cellular ceramide pool can be generated as a result of three metabolic activities: (1) the de novo synthesis pathway, (2) the hydrolysis of sphingolipids, or (3) the ceramide synthase activity in a salvage pathway [61]. The de novo biosynthesis of ceramide occurs in the endoplasmic reticulum, and its first and rate-controlling step is catalyzed by serine palmitoyltransferase, which condenses preferentially serine and palmitoyl-CoA to form 3-ketodihydrosphingosine. The degradation of sphingolipids takes place mainly in the acidic late endosomes and lysosomes. In brief, glycosidases’ stepwise removal of the oligosaccharide chain leads to ceramide generation. Moreover, sphingomyelin can be converted to ceramide by acid sphingomyelinase. Several studies demonstrated rapid activation of acid sphingomyelinase and neutral sphingomyelinase in cultured cells [62,63] as well as in vivo (rat lung epithelium) upon LPS challenge [64].

In the study, HMC3 treatment with PA combined with LPS reversed characteristic changes in lipid content generated by LPS, leading to similar metabolic profiles of PA and PA+LPS samples. For instance, sphingomyelin content was significantly lower in PA and PA+LPS-treated cells than in LPS-treated or control cells (exception: SM d18:1/16:0—no difference observed). Also, ceramide species in PA and PA+LPS-treated cells had lower concentrations than in LPS-challenged HMC3 samples. According to Lu et al. [65], PA and PA+LPS stimulated ceramide production in HMC3 cells by increasing both ceramides’ synthesis de novo and via sphingomyelins hydrolysis. Similarly, in our study, in pro-inflammatory stimulated HMC3 cells, PA combined with LPS and PA individually, but not LPS alone, decreased levels of SM(d18:1/14:0) and SM(d18:1/18:0), suggesting PA-induced SM hydrolysis. According to Józefowski et al. [63], in mouse macrophages, LPS at very low concentrations (1–2 ng/mL) activated sphingomyelinases and moderately increased the level of ceramides. However, this effect was inhibited, among others, by inhibitors of endotoxin-induced TNFα. Moreover, exposure of macrophages to bacterial sphingomyelinases or inhibition of ceramide hydrolysis by ceramidase inhibitors reduced the production of TNFα. This suggests that ceramides negatively regulate the production of TNFα in response to LPS and thus may have anti-inflammatory effects, which in our studies seem to be confirmed by increased expression of TREM2 and TGFβ secretion and lack of IL-1β production. On the other hand, PA in our study decreased ceramide levels and TREM2 expression in HMC3 cells, thus opposingly affecting ceramide production and inhibiting inflammation.

On the other hand, one consequence of TLR4-mediated macrophage activation is the remodeling of lipids, leading to increased total cellular sphingolipids, including ceramides [66]. Shilling et al. [66] on mouse primary macrophages showed that TLR4 stimulation with LPS generated ceramide via de novo synthesis, among others. Moreover, LPS at low concentrations synergized with PA to increase ceramide C16 and C16 ceramide phosphate. However, our results did not show significant differences in ceramide C16 levels in HMC3 cells challenged with LPS and PA. In contrast, LPS increased long-ceramide chain ceramide content in HMC3 cells. Whether LPS-induced TLR4 signaling impacts ceramide C18 synthesis in inflammatory-activated microglial cells must be elucidated all the more so because the level of C18 ceramide is increased in the cerebrospinal fluid of patients with AD and positively correlated with T-tau among the cohort of patients at risk of or at the early stages of dementia, suggesting its role in neurodegeneration [67].

Furthermore, it has been established that PA can also affect TLR4 signaling pathways, and we could observe that the order of stimulation of HMC3 cells with LPS and PA also affects TREM2 expression (preliminary results not shown here). Treatment of HMC3 first with PA and one hour later with LPS reduced TREM2 expression in endotoxin—challenged cells, suggesting competition between LPS and PA for the TLR4 receptor. However, further studies are needed to determine how these pro-inflammatory stimuli alter microglial lipid metabolism via the receptor.

Provision of 200 µM PA (alone and with LPS) to the HMC3 cells induced accumulation of TAGs regardless of the length of their fatty acids, but not ceramides of SMs. This suggests supply-induced partitioning of palmitate towards diacylglycerides (DAG) and TAG synthesis rather than palmitoyl-CoA used by serine palmitoyltransferase. Moreover, these conditions not only lead to increased synthesis of palmitic acid-containing TAGs (e.g., tripalmitin: TAG 16:0/16:0/16:0) but might have affected the diminished catabolism of unsaturated long-chain fatty acids, as the relative increase in the content of lipids composed of those fatty acids was also observed. SFAs transported to cells are converted into fatty acyl-CoAs, which may be then catabolized to generate energy or directed to produce DAGs, subsequently converted to TAGs, and incorporated into lipid droplets (LDs) thus providing storage of neutral lipids [68,69]. On the one hand, it is assumed that LDs, which play a central role in lipid metabolism, protect cells from stress, including lipotoxic stress [70]. Cheon et al. [71] showed that converting PA in muscle cells to neutral TAGs reduces its lipotoxicity. Whether the metabolism of PA to TAGs plays a similar role in microglia remains to be determined. However, it has also been proven that LDs accumulating in microglia with aging disrupt its functions, which manifest in reduced phagocytosis, oxidative stress, and the secretion of pro-inflammatory cytokines [72]. It has been proven that LDs are also associated with neurodegeneration; thus, the effect of LDs on microglia is likely dependent on the quantity and quality of lipids deposited in them [19].

The microglial lipidome drastically changes during inflammation, expressing as upregulation of lipid-related genes, accumulation of LDs, compositional remodeling of membrane phospholipids, and conformational changes of membrane proteins [19]. Hence, knowledge of lipid metabolism in microglia, especially in chronic inflammation accompanying ME and obesity, is crucial for developing effective methods of preventing and treating neurodegeneration. Although much is already known about the role of lipids in neurodegenerative diseases, this knowledge still needs to be revised. At the same time, research is hampered by the fact that lipid metabolism in the brain is complex and multifactorial.

Our study has several limitations, as do all in vitro studies on microglial cells, reflecting only a fraction of the complex interactions between metabolism, inflammation, and microglia activating factors. First, the analysis of the lipid profile in HMC3 microglial cells challenged with PA and LPS showed that the BSA we used for conjugation with PA was contaminated with endotoxin, as indicated by similar lipid profiles for the vehicle control with BSA and LPS. Contamination of BSA with endotoxin is a common problem in vitro studies of the effect of saturated fatty acids on cells, as described by Yang et al. [73] and Cullberg et al. [74]. Similarly, different conjugation protocols of PA with BSA and various types of alcohol used to dissolve PA may affect the results obtained. Hence, in the study, only significant differences in the levels of lipids analyzed were considered. Next, we analyzed only a few cytokines and no inflammatory-related receptor expression, which makes it challenging to analyze research results.

Another limitation of our study is that we did not have a control group of TNFα-unstimulated HMC3 cells. This made it impossible to compare the effect of this cytokine on the lipid profile and secretion of the studied cytokines by microglial cells under the experimental conditions used.

Nevertheless, our studies indicate a different response of microglia to varying configurations of inflammatory stimulators. The effect of TNFα on microglia interactions with endotoxin and SFAs seems particularly important and certainly requires further, detailed studies. The potential impact of pre-stimulation of human microglia cells, probably through the activation of TNFα inhibitors, in inhibiting the inflammation induced by LPS and PA but affecting lipid metabolism, underscores the urgency and importance of further detailed studies.

In our studies, in contrast to most previously published results, LPS, but not PA, induced an increase in ceramides and TREM2 expression in microglia, which requires confirmation in another cell model and in vivo animal studies. The use of prolonged exposure of microglia to three potent pro-inflammatory stimulants, i.e., LPS, PA, and TNFα, showed that PA can be metabolized to LDs, which, taking into account many months of endotoxemia and systemic inflammation accompanying obesity and ME, may ultimately lead to neurodegeneration, similarly to the LPS-induced increase in ceramides.

## 5. Conclusions

(1) Prior activation of HMC3 with TNFα cells, then stimulated with LPS and PA, silenced the microglial immune activation but changed lipid metabolism;

(2) LPS enhanced, while PA reduced TREM2 expression in TNFα-activated microglial cells;

(3) LPS increased ceramide content, while PA caused TAG accumulation in HMC3 cells;

(4) TNFα-activated microglial cells exposed to LPS and PA showed a propensity to a neurodegenerative phenotype.

Altogether, our results indicated that pre-stimulation of microglia with TNFα significantly affected its interactions with LPS and PA and changed lipid metabolism, which may lead to the silencing of microglial activation and neurodegeneration.

## Figures and Tables

**Figure 1 biomolecules-14-01238-f001:**
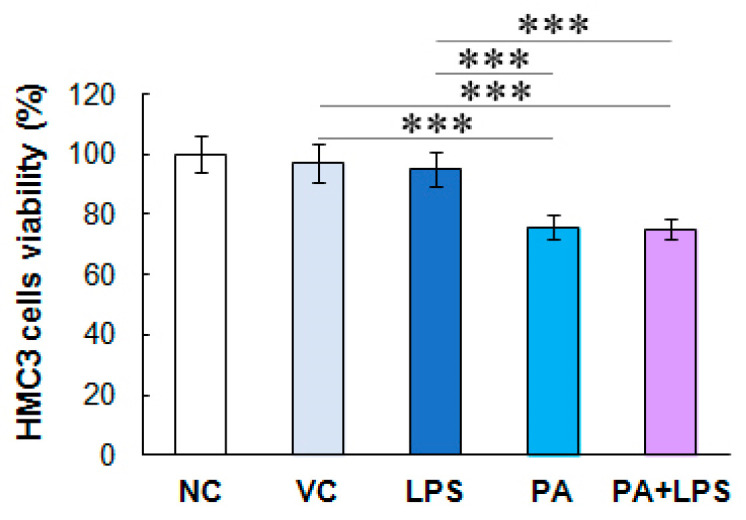
HMC3 cell viability. The microglial cell viability was assessed after treatment with LPS, PA, and LPS+PA. The cells were pre-stimulated for 24 h with 25 ng/mL TNFα and then treated for 24 h with LPS (10 ng/mL) or PA (200 µM) alone, or both. NC, negative control; VC, vehicle control. The results are the average of two biological replicates, each with eight samples (*n* = 16). *** *p* < 0.0001 indicates significant differences between untreated and LPS, PA, or LPS+PA-treated cells.

**Figure 2 biomolecules-14-01238-f002:**
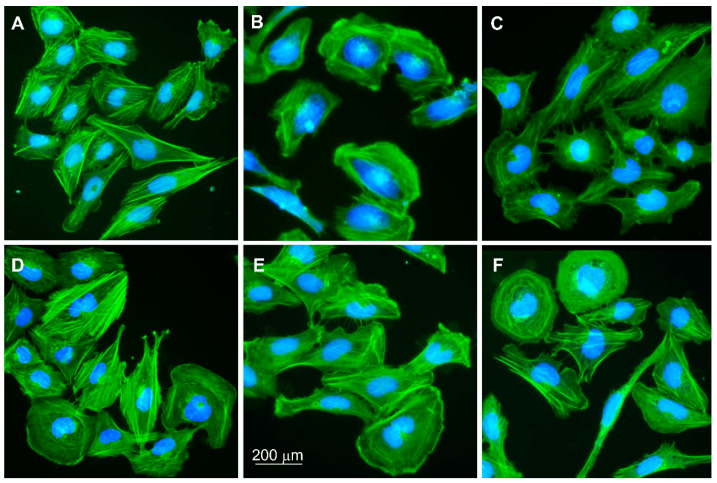
HMC3 cell morphology. (**A**) Untreated microglial cells; HMC3 stimulated with (**B**) TNFα at 25 ng/mL for 24 h—negative control (NC) in the study; (**C**) VC, vehicle control (BSA-NaCl); (**D**) LPS at 10 ng/mL; (**E**) PA at 200 mM; (**F**) LPS+PA. HMC3 cells were stained with phalloidin-FITC and DAPI. Fluorescence microscope; magnification 200×.

**Figure 3 biomolecules-14-01238-f003:**
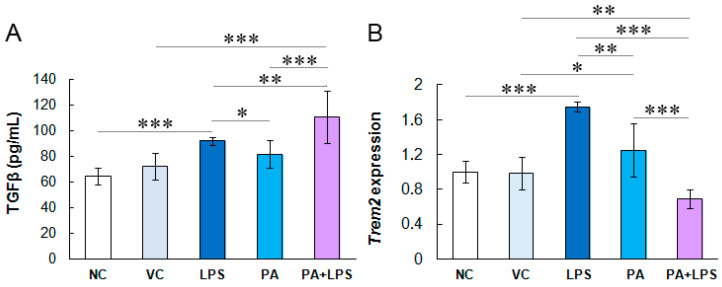
(**A**) Anti-inflammatory TGFβ cytokine production by HMC3 cells. The cells were pre-stimulated for 24 h with 25 ng/mL TNFα and then treated for 24 h with LPS (10 ng/mL), PA (200 µM), or both. (**B**) Fold-change of *Trem2* gene expression in HMC3 cells. NC, negative control; VC, vehicle control. The results are the average of three biological replicates, each with three samples (*n* = 9). *** *p* < 0.0001, ** *p* < 0.001, and * *p* < 0.05 indicates statistically significant differences between untreated and LPS, PA, or LPS+PA-treated cells.

**Figure 4 biomolecules-14-01238-f004:**
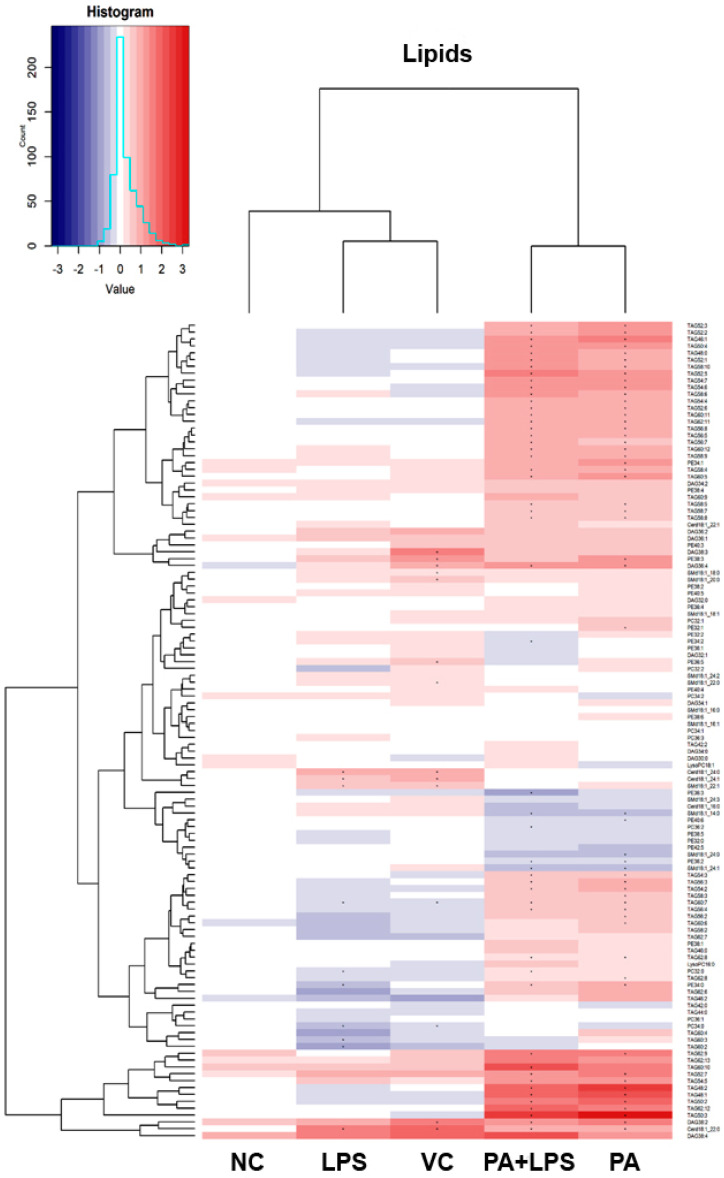
Heatmap of identified lipids. The heatmap visualizes normalized and scaled chromatographic peak areas as Log_2_ ratios of treatment to control. The cells were pre-stimulated for 24 h with 25 ng/mL TNFα and then treated for 24 h with LPS (10 ng/mL), PA (200 µM), or LPS+PA. NC, negative control; VC, vehicle control. Asterix (*) indicates a statistically significant (*p* < 0.05) difference from the control.

**Figure 5 biomolecules-14-01238-f005:**
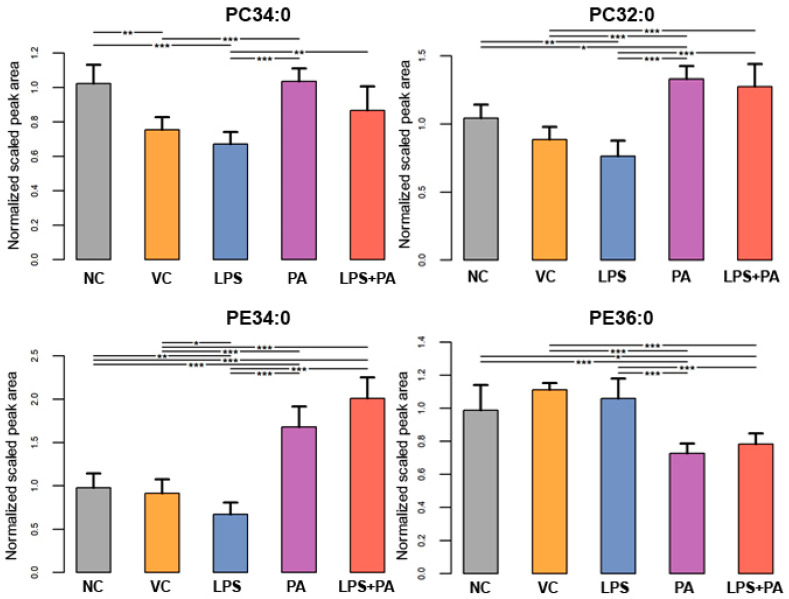
Lipidomics analysis results for phospholipids. The cells were pre-stimulated for 24 h with 25 ng/mL TNFα and then treated for 24 h with LPS (10 ng/mL), PA (200 µM), or LPS+PA. NC, negative control; VC, vehicle control. *** *p* < 0.0001, ** *p* < 0.001, and * *p* < 0.05 indicates statistically significant differences between untreated and LPS, PA, or LPS+PA-treated cells.

**Figure 6 biomolecules-14-01238-f006:**
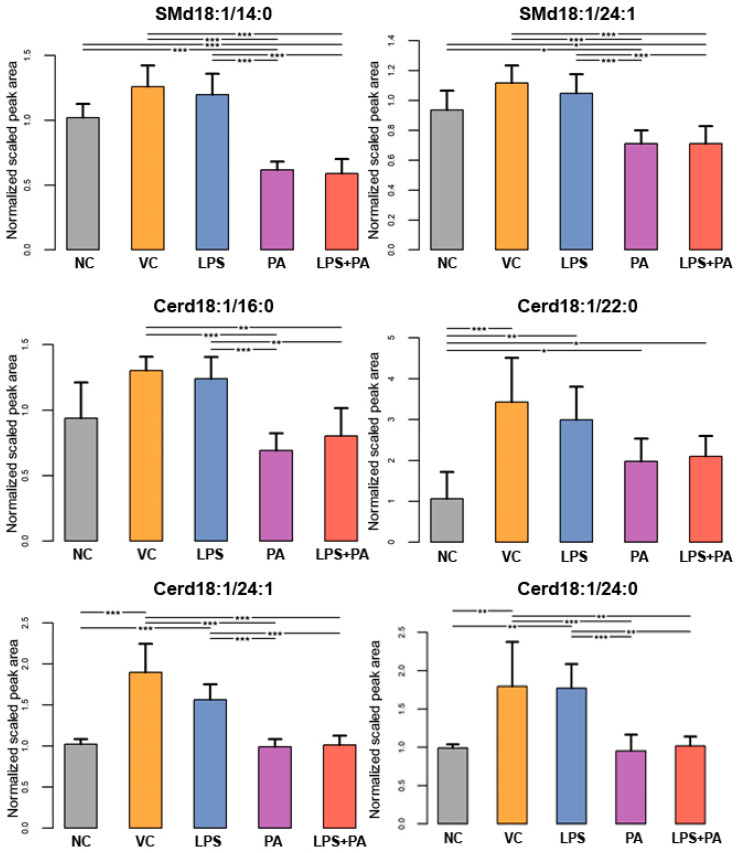
Lipidomics analysis results for sphingolipids and ceramides. The cells were pre-stimulated for 24 h with 25 ng/mL TNFα and then treated for 24 h with LPS (10 ng/mL), PA (200 µM), or LPS+PA. NC, negative control; VC, vehicle control. *** *p* < 0.0001, ** *p* < 0.001, and * *p* < 0.05 indicates statistically significant differences between untreated and LPS, PA, or LPS+PA-treated cells.

**Figure 7 biomolecules-14-01238-f007:**
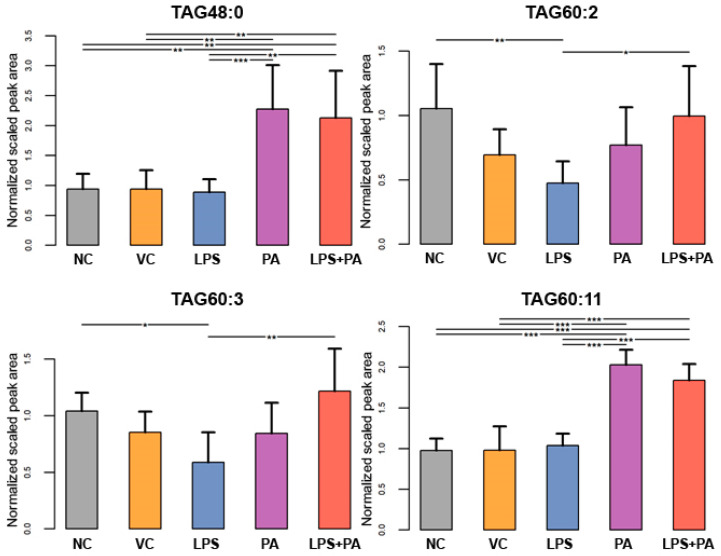
Lipidomics analysis results for triacylglycerides (TGA). The cells were pre-stimulated for 24 h with 25 ng/mL TNFα and then treated for 24 h with LPS (10 ng/mL), PA (200 µM), or LPS+PA. NC, negative control; VC, vehicle control. *** *p* < 0.0001, ** *p* < 0.001, and * *p* < 0.05 indicates statistically significant differences between untreated and LPS, PA, or LPS+PA-treated cells.

## Data Availability

Data are available on demands in correspondence author.

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
