# Peer review of "Lipidomics Analysis of Human HMC3 Microglial Cells in an In Vitro Model of Metabolic Syndrome"

_biomolecules, 2024, doi:10.3390/biom14101238_

Round 1
Reviewer 1 Report
Comments and Suggestions for Authors
Overview of the paper:
The study explores how pre-stimulation of microglia with TNFα influences their response to endotoxin (LPS) and palmitic acid, affecting lipid metabolism and immune response. The study finds that while LPS increases TREM2 expression and ceramide levels, palmitic acid decreases TREM2 expression and increases triglyceride levels, potentially contributing to neurodegeneration.
The authors have done an excellent job, particularly in the discussion section, which is well-articulated and thoroughly compared with published studies. The following suggestions are minor and intended to improve the readability of the document. However, I believe adding a NOS/ROS measurement experiment could significantly enhance the paper.
Material and Methods:
- The authors describe how the culture of HMC3 was done, but it is missing how cells were kept in the long term. How often were medium changes performed? When cells reached confluency, what happened? Please update the methodology description accordingly.
- Line 110, what do the authors mean by PA was complexes? Maybe the authors want to say mixed?
- In section 2.3, which kind of plastic were the cells in? Can the authors re-write this section to be clearer?
- Lines 139 to 143, the authors should re-write this part; it is not clear what they have done.
- Line 154: ΔΔCt method would make sense to have a reference for this method or describe it briefly what it does.
- 215- Spelling of TNF-a
Results/Discussion:
On Figures 4,5, 6: I would recommend increasing the graph sizes, together with the statistics, for better readability.
317 line: The Blood Brain Barrier abbreviation was already used in the introduction. I found some more like that as well. I would recommend organizing the manuscript in terms of abbreviations and not repeating what was done.
The authors should consider adding a NOS/ROS measurement to have another assay for microglia activation.
Comments on the Quality of English LanguageThe authors should look for some grammatical errors, and clarity as I pointed out above.
Author Response
Overview of the paper: The study explores how pre-stimulation of microglia with TNFα influences their response to endotoxin (LPS) and palmitic acid, affecting lipid metabolism and immune response. The study finds that while LPS increases TREM2 expression and ceramide levels, palmitic acid decreases TREM2 expression and increases triglyceride levels, potentially contributing to neurodegeneration. The authors have done an excellent job, particularly in the discussion section, which is well-articulated and thoroughly compared with published studies.
The following suggestions are minor and intended to improve the readability of the document.
- However, I believe adding a NOS/ROS measurement experiment could significantly enhance the paper.
Answer: Unfortunately, the HMC3 cell line used for the study does not produce detectable amounts of NO (we have checked it several times in the Griess reaction). Since we have no grant for the research and only minor financial support, we could not perform additional assays.
Material and Methods:
- The authors describe how the culture of HMC3 was done, but it is missing how cells were kept in the long term. How often were medium changes performed? When cells reached confluency, what happened? Please update the methodology description accordingly.
Answer: Information on HMC3 cell culture has been completed (highlighted in yellow).
- Line 110, what do the authors mean by PA was complexes? Maybe the authors want to say mixed?
Answer: We meant complexed as combined with BSA; for clarity, we have changed the word from ‘complexed’ to ‘combined.’
- In section 2.3, which kind of plastic were the cells in? Can the authors re-write this section to be clearer?
Answer: The information has been expanded and is highlighted in yellow.
- Lines 139 to 143, the authors should re-write this part; it is not clear what they have done.
Answer: We have re-write this paragraph, and all changes are highlighted in yellow.
- Line 154: ΔΔCt method would make sense to have a reference for this method or describe it briefly what it does.
Answer: We have added the reference to the method.
- 215- Spelling of TNF-a
Answer: Thank you very much; that error has been corrected.
Results/Discussion:
- On Figures 4,5, 6: I would recommend increasing the graph sizes, together with the statistics, for better readability.
Answer: Unfortunately, the program generated these figures for lipidomics analysis, and we cannot change them. However, we have attached all Figures in tif (600 dpi) with the hope that it will improve their readability.
- 317 line: The Blood Brain Barrier abbreviation was already used in the introduction. I found some more like that as well. I would recommend organizing the manuscript in terms of abbreviations and not repeating what was done.
Answer: All abbreviations were developed when cited the first time. The whole manuscript was checked for unnecessary repeats of developed abbreviations.
- The authors should consider adding a NOS/ROS measurement to have another assay for microglia activation.
Answer: Unfortunately, the HMC3 cell line we used in our study does not produce NO (we have checked this several times with the Griess reaction). Also, as we did not have financial support for our study, we were not able to perform more assays for microglia activation.
- The authors should look for some grammatical errors, and clarity as I pointed out above.
Answer: The whole manuscript was checked carefully for any errors.
Reviewer 2 Report
Comments and Suggestions for Authors
- A brief summary
The authors did the cytokine ELISA and Lipidomic analysis on the TNFα pre-stimulated and LPS&palmitic acid stimulated HMC3 cells.
- General comments
1. The novelty of this study comes from the TNFα pre-stimulated HMC3 cells. However, the study does not include TNFα negative cells as a critical control.
2. The heatmap in the supplementary materials is a better data visualization that demonstrates the high levels throughout the Lipidomic dataset. Figures 4, 5, and 6 still give less information than the heatmap.
3. In the abstract: Line 20, “pre-stimulated with TNFα conflicts with line 22, “stimulated for 24 h with TNFβ”.
4. For Cytokine production assays, which commercial kit was used? Please add the catalog number for all the commercial chemicals or kits used in this study. It’s not convincing that the kit cannot detect the IL-1β, IL-10, and IL-13 in microglia-like cell lines after LPS treatment.
- Specific comments
1. Line 130: 20 ng/mL tumor necrosis factor α (TNFα) for 24 h to activate microglial cells, which conflict with the description in the supplementary materials: pre-stimulated for 24 h with 25 ng/mL TNFα and then treated for 24 h.
2. What’s the difference between the two heatmaps in the supplementary materials?
3. Add the group size to the figure legends.
4. Label the treatment or group information in Figure 2.
5. The error bar of the PA group was cropped in Figure 3A. The error bar of the VC group is not symmetrical. It’s hard to believe the p-value between the LPS group and the PA +LPS group is smaller than 0.001 with such a big error bar.
6. What is the mean of the y-axis in Figures 4, 5, and 6?
Author Response
The authors did the cytokine ELISA and Lipidomic analysis on the TNFα pre-stimulated and LPS&palmitic acid stimulated HMC3 cells.
General comments
- The novelty of this study comes from the TNFα pre-stimulated HMC3 cells. However, the study does not include TNFα negative cells as a critical control.
Answer: Thank you for your valuable comment. Unfortunately, due to the lack of funding for our studies, we had to limit the number of samples tested for lipidomics to perform three biological repeats.
- The heatmap in the supplementary materials is a better data visualization that demonstrates the high levels throughout the Lipidomic dataset. Figures 4, 5, and 6 still give less information than the heatmap.
Answer: We added the heatmap to the manuscript according to your suggestion.
- In the abstract: Line 20, “pre-stimulated with TNFα conflicts with line 22, “stimulated for 24 h with TNFβ”.
Answer: This mistake has been corrected.
- For Cytokine production assays, which commercial kit was used? Please add the catalog number for all the commercial chemicals or kits used in this study. It’s not convincing that the kit cannot detect the IL-1β, IL-10, and IL-13 in microglia-like cell lines after LPS treatment.
Answer: The revised version of our manuscript includes all necessary information concerning ELISA kits. To detect these cytokines, we selected kits that detect the lowest concentrations of the assayed cytokines (changes are highlighted in yellow in the manuscript).
Specific comments
- Line 130: 20 ng/mL tumor necrosis factor α (TNFα) for 24 h to activate microglial cells, which conflict with the description in the supplementary materials: prestimulated for 24 h with 25 ng/mL TNFα and then treated for 24 h.
Answer: That was an error, which has been corrected in the revised version of the manuscript. HMC3 cells were stimulated with 25 ng/mL of TNFα.
- What’s the difference between the two heatmaps in the supplementary materials?
Answer: Both heatmaps presented all 1483 chromatographic features in our experiment and a selected subset of identified 147 lipids. As these were discussed in detail in the manuscript, the illustration of the subset is now part of the manuscript as Fig. 4. Both figures received clear captions as well.
- Add the group size to the figure legends.
Answer: The group size has been added to the figure legends.
- Label the treatment or group information in Figure 2.
Answer: The legend of Figure 2 has been corrected.
- The error bar of the PA group was cropped in Figure 3A. The error bar of the VC group is not symmetrical. It’s hard to believe the p-value between the LPS group and the PA +LPS group is smaller than 0.001 with such a big error bar.
Answer: Thank you very much for the comment. We made a mistake; SD values ​​were entered incorrectly in the graphs (decimal point shifted). This error has been corrected.
- What is the mean of the y-axis in Figures 4, 5, and 6?
Answer: Figures 4,5,6 (old numbering) depicts the mean normalized, scaled peak area of three biological replicates, each with a mean of 3 technical replicates. Peak areas in each sample were normalized to a sample's internal standard area to accommodate any losses in extraction. Next, each analyte's normalized peak areas were scaled - divided by the mean of the control sample. Thus, bar plots represent the amount of an analyte in treatment samples relative to the control–negative control for LPS-treated cells and vehicle control for cells treated with PA and LPS+PA. The designation of the y-axis has been added to all figures in the revised version of the manuscript.
Reviewer 3 Report
Comments and Suggestions for Authors
the work by Mateusz Chmielarz et al. titled “Lipidomics analysis of human HMC3 microglial cells in an in vitro model of metabolic syndrome” investigates obesity-associated systemic inflammation linked to bacterial endotoxin (LPS) and elevated saturated fatty acids. In microglia pre-stimulated with TNFα, LPS increased TREM2 expression and ceramide levels, while palmitic acid (PA) reduced TREM2 and increased triglycerides. These changes in lipid metabolism, as suggested by the authors, may lead to reduced microglial activation and neurodegeneration. I found the work very interesting
Why is it claimed that reduced microglial activation leads to neurodegeneration? It is known from the literature that chronic overactivation of microglia in a pro-inflammatory sense leads to chronic inflammation with accumulation of ROS increased secretion of inflammation mediators leading to neuronal death.
An assay highlighting the number of live and dead cells after the various treatments (e.g., Trypan blue assay), which is important for assessing cytotoxicity, was not included in the cell viability assay.
Immunoassays used for quantification of cytokines in the culture medium were not specified in the materials and methods.
In addition, enzyme immunoassays have different sensitivities, and detecting cytokine production in the culture medium can be complex. Why wasn't RT-PCR performed on cell lysates to evaluate the mRNA of the indicated cytokines?
In the immunofluorescence image, to make the differences between treatments more usable, it would be helpful to add a graph.
Also, I would like a clarification: already treatment with only TNF-alpha (pro-inflammatory cytokine) should lead to an ameboid-like microglial phenotype with fewer elongations, in this case it seems to me that the opposite is stated.
Author Response
The work by Mateusz Chmielarz et al. titled “Lipidomics analysis of human HMC3 microglial cells in an in vitro model of metabolic syndrome” investigates obesity-associated systemic inflammation linked to bacterial endotoxin (LPS) and elevated saturated fatty acids. In microglia pre-stimulated with TNFα, LPS increased TREM2 expression and ceramide levels, while palmitic acid (PA) reduced TREM2 and increased triglycerides.
These changes in lipid metabolism, as suggested by the authors, may lead to reduced microglial activation and neurodegeneration. I found the work very interesting.
- Why is it claimed that reduced microglial activation leads to neurodegeneration? It is known from the literature that chronic overactivation of microglia in a pro-inflammatory sense leads to chronic inflammation with accumulation of ROS increased secretion of inflammation mediators leading to neuronal death.
Answer: Reduction of microglial activation, associated with, among others, decreased phagocytic activity, affects many microglial functions, such as neuronal pruning, phagocytosis of cell debris, or amyloid deposits. Hence, both excessive activation and attenuation of microglial activity may affect the development of neurodegenerative diseases.
- An assay highlighting the number of live and dead cells after the various treatments (e.g., Trypan blue assay), which is important for assessing cytotoxicity, was not included in the cell viability assay.
Answer: That is true. We have not performed the Trypan Blue Assay. However, I must mention that the MTT assay provides more information regarding cell viability as it measures both viability and cells' metabolic/mitochondrial activity. According to our knowledge, the MTT assay gives more accurate results than the Trypan Blue one. Hence, we used the MTT assay.
- Immunoassays used for quantification of cytokines in the culture medium were not specified in the materials and methods. In addition, enzyme immunoassays have different sensitivities, and detecting cytokine production in the culture medium can be complex.
Answer: The immunoassays we used are specified in the revised version of our manuscript. Paragraph 2.8, lines 220-222, highlighted in yellow.
- Why wasn't RT-PCR performed on cell lysates to evaluate the mRNA of the indicated cytokines?
Answer: Unfortunately, despite our best efforts for two years, we could not obtain a research grant, so we had no financial support and could not carry out the planned research. However, despite many shortcomings, we hope our publication will find enough recognition to be published.
- In the immunofluorescence image, to make the differences between treatments more usable, it would be helpful to add a graph.
Answer: We stained the actin cytoskeleton of cells to visualize cell morphology, so the fluorescence intensity is not significant in this fluorescent staining to make graphs.
5.. Also, I would like a clarification: already treatment with only TNFalpha (pro-inflammatory cytokine) should lead to an ameboid-like microglial phenotype with fewer elongations, in this case it seems to me that the opposite is stated.
Answer: That is true, and cells were rounded after treatment with TNFα only (Figure 2B) compared to untreated cells (Figure 2A). However, pre-stimulation with TNFα and then treatment with LPS and/or PA influenced cell morphology into more elongated cells, although there were also round cells. These differences in cell morphology most probably reflected their decreased phagocytic activity. Hence, our next step is to apply for a grant and study these processes in detail.
Round 2
Reviewer 2 Report
Comments and Suggestions for Authors
I don't believe that the HMC3 cell doesn't produce detectable amounts or secrete negligible amounts of the pro-inflammatory cytokine IL1β. The HMC3 is a microglia-like cell and was chosen as a model to study the inflammatory response. Researchers (https://doi.org/10.3892/etm.2024.12456) have already tested the IL1β level by ELISA; the background IL1β level is already higher than the threshold of the kits used by the authors. I recommended to reject this manuscript. secreted negligible amounts of the pro-inflammatory cytokine IL-1β
Author Response
Dear Reviewer,
Thank you for your feedback. As suggested, in the revised version of the manuscript, we have included a reference to the lack of unstimulated TNFα control (Discussion, lines 599-601) and the IL-1b results, referring to the results of the studies presented by Chen et al. [2024] in the publication on 'Modulation of secretory factors by lipofundin contributes to its anti‑neuroinflammatory effects' (Discussion, lines 386-395). The fragments added in the revised version of our manuscript are highlighted in yellow.
We hope that reviewer will consider the corrections made, which significantly improved our manuscript, and that the revised version will meet the requirements for its publication in Biomolecules.